# Evolution from Periodic Intensity Modulations to Dissipative Vector Solitons in A Single-Mode Fiber Laser

**Xiao Hu [1,2], Jun Guo [1,*], Lei Li [2], Seongwoo Yoo [2] and Dingyuan Tang [1,2]**

[1]  Jiangsu Key Laboratory of Laser Materials and Devices, School of Physics and Electronic Engineering, Jiangsu Normal University, Xuzhou 221116, China; xiao003@e.ntu.edu.sg (X.H.); edytang@ntu.edu.sg (D.T.)

[2]  School of Electrical and Electronic Engineering, Nanyang Technological University, Singapore 639798, Singapore; leelei@jsnu.edu.cn (L.L.); seon.yoo@ntu.edu.sg (S.Y.)

[*]  Correspondence: guojun@jsnu.edu.cn; Tel.: +86-130-1616-2170

**Abstract:** We investigated—both experimentally and numerically—the operation of a weakly birefringent cavity fiber laser under different net cavity dispersion values. Experimentally, we found that under coherent cross-polarization coupling, either in-phase or anti-phase low frequency intensity modulations between the two orthogonal polarization components of the laser emission could be obtained. The evolution of the periodic intensity modulations in the fiber laser under different operation conditions was studied. In this paper, we show that under suitable conditions, they can be shaped into a train of bright-bright, dark-dark, or dark-bright vector solitons.

**Keywords:** fiber lasers; nonlinear optics; optical solitons

## 1. Introduction

When two or more optical fields co-propagate in an optical fiber, they will couple with each other through the fiber nonlinearity. The most common mode of coupling is cross-phase modulation (XPM), which refers to the nonlinear changes of an optical field induced by the copropagating fields [1]. It has been shown that under cross-phase modulation between lights, various new effects can occur in a single-mode fiber (SMF) [2–6]. It is worth mentioning that, in 1987, Agrawal theoretically predicted that, through incoherent coupling between two or more optical fields, modulation instability (MI) could even occur in a normal dispersion fiber [7], despite the fact that conventional MI could only occur in an anomalous dispersion fiber [8]. In addition, MI caused by cross-phase modulation has also been observed in fiber amplifiers and fiber lasers [9–13].

It is well-known that, due to the existence of fiber birefringence, an SMF intrinsically supports two orthogonal polarization modes. Therefore, in practice, the light propagation in SMFs can involve coupling between the two modes. Depending on the strength of fiber birefringence, the two modes can either be incoherently or coherently coupled [1]. In 1988, Wabnitz theoretically predicted a kind of polarization modulation instability (PMI) [14]. He showed that, due to the coherent cross-phase modulation between two circularly polarized modes in a weakly birefringent SMF, MI could occur in both the normal and anomalous dispersion regimes. In particular, unlike conventional MI or MI that occurred under incoherent cross-polarization coupling, where the MI gain peaks were always located at a very high modulation frequency [8,15–18], under coherent cross-polarization coupling, the modulation instability gain peaks could be situated at a low or even vanishing modulation frequency. Later, Chiu and Chow theoretically showed that PMI could occur under coupling between two orthogonal linearly polarized modes and a finite birefringence of the fiber could even enhance the PMI [19].

A fiber laser is an excellent testbed for the experimental study of nonlinear fiber optic effects as, under suitable conditions, its dynamics mimics those of light propagation in fibers. Depending on the laser configuration and operation conditions, different nonlinear fiber optical effects, such as MI [10,11] and bright and dark soliton formation [20,21], have been observed and extensively investigated in fiber lasers. Recently, in quasi-isotropic cavity fiber lasers, different types of vector solitons, such as bright-bright [22], dark-dark [23], and dark-bright vector solitons [24], were also experimentally observed. Different from the conventional phase-locked bright-bright vector solitons formed in mode locked fiber lasers [25,26], the vector solitons could automatically appear in a fiber laser without inserting a saturable absorber or mode locker into the cavity. To determine the formation mechanisms of the vector solitons, we investigated—both experimentally and numerically—the formation procedure of the vector solitons in the fiber lasers. In this paper, we report our research results. The paper is organized as follows: In Section 2, we report on the experimental studies on vector soliton formation in the fiber lasers; Section 3 presents the numerical simulation results; Section 4 discusses the possible mechanisms of the variable vector soliton formation observed in the fiber lasers; and Section 5 presents the conclusion.

## 2. Experimental Details

We first constructed an all-normal dispersion cavity fiber ring laser, as schematically shown in Figure 1. It had a ring cavity consisting of 3 m erbium-doped fiber (OFS-EDF80) with a group velocity dispersion (GVD) coefficient of 61.7 ps$^2$/km and 38.3 m dispersion compensation fiber (DCF) with a GVD coefficient of 5.1 ps$^2$/km. The fiber laser was pumped by a 1480 nm SMF Raman laser, whose maximum output power was as large as ~5 W. A polarization-independent isolator (ISO) was inserted into the cavity to force the unidirectional circulation of the cavity. In addition, an intra-cavity polarization controller (PC) was used to fine-tune the linear cavity birefringence. A wavelength division multiplexer (WDM) was used to couple the pump light in the cavity, and a 10% fiber coupler was used to output the light. All the intracavity components (WDM, ISO, and PC) were made or pigtailed with the DCF. No real or artificial saturable absorber was inserted into the cavity. Experimentally, to separate the two orthogonal polarization components of the laser emission, the laser output was first sent to a fiber pigtailed polarization beam splitter and then monitored with a high-speed electronic detection system consisting of two 40-GHz photodetectors and a 33-GHz bandwidth real-time oscilloscope (Agilent Technologies Singapore, DSA-93204A). A polarization controller was inserted between the laser output and the polarization beam splitter to balance the linear polarization change caused by the lead fibers. An optical spectrum analyzer (Yokogawa, AQ6375) was used in our experiment to monitor the optical spectrum of the laser emission.

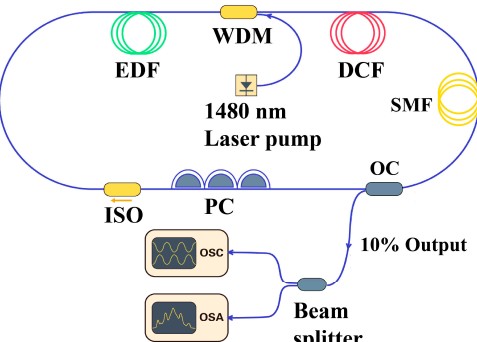

**Figure 1.** A schematic of the Er-doped fiber (EDF) ring laser. DCF, dispersion-compensation fiber; WDM, wavelength division multiplexer; SMF, single-mode fiber; PC, polarization controller; ISO, isolator; OC, output coupler; OSC, oscilloscope; OSA, optical spectrum analyzer.

While building the fiber cavity, special care was taken to ensure that the net cavity birefringence was sufficiently small so that the laser could simultaneously oscillate in its two orthogonal linear polarization modes. Strong coherent XPM between the two lasing modes could be achieved in the laser by carefully tuning the intracavity PC paddles. Experimentally, we used the wavelength separation between the two lasing modes as an indicator of the net cavity birefringence. Under coherent cross-polarization coupling, the central wavelengths of the laser emissions along the two orthogonal polarization directions virtually overlapped.

The fiber laser had an average cavity GVD coefficient $\beta_2 \approx 9.7 \ \mathrm{ps}^2/\mathrm{km}$. When turning the pump power up, above the laser threshold, the laser simultaneously emitted Continuous wave (CW) along its two orthogonal cavity polarization directions. By increasing the pump power to 30 dBm, at an appropriate setting of the PC paddles, the state shown in Figure 2 was obtained. Figure 2a shows the polarization resolved laser emissions along the two orthogonal polarization directions of the cavity. The intensity of the polarization resolved laser emissions was in-phase periodically modulated. In the case shown, it had a modulation frequency of 400 MHz. Other low frequency periodic intensity modulations were also experimentally obtained. The low frequency periodic intensity modulation only appeared under coherent cross-polarization coupling and always simultaneously occurred on the two orthogonal polarization components of the laser emission.

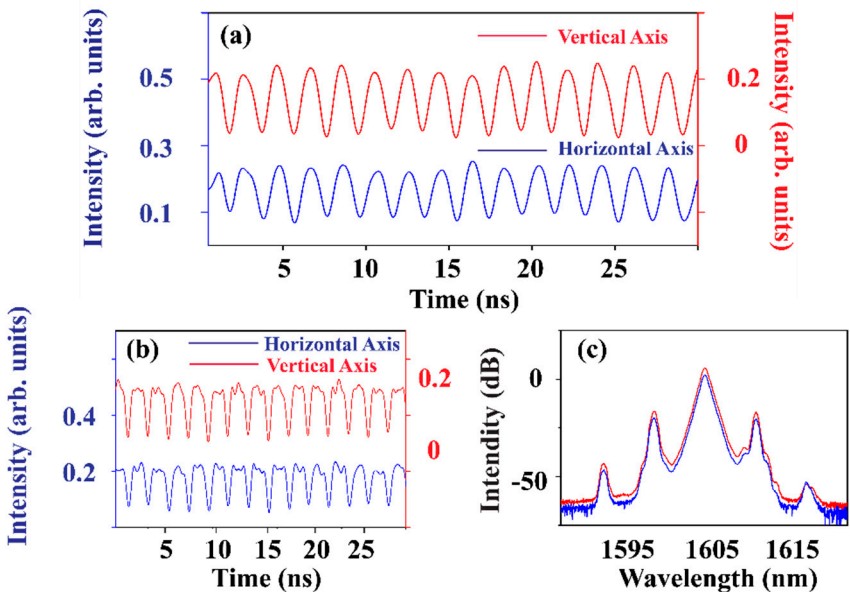

**Figure 2.** Evolution of in-phase periodic intensity modulations in the fiber laser (erbium-doped fiber (EDF): 3 m; single-mode fiber (SMF): 0 m; dispersion compensation fiber (DCF): 38 m). (**a**) Initial in-phase periodic intensity modulation; (**b**) dark-dark pulse pairs formed; (**c**) corresponding polarization resolved optical spectra of (**b**).

Starting from such a low frequency in-phase intensity modulation state, when the pump power was further increased to 33 dBm, the state shown in Figure 2b could be obtained. The in-phase periodic intensity modulation evolved into a stable train of coupled dark-dark pulse pairs. The dark pulses had a pulse width of around 1 ns, which was much broader than those expected for the dark solitons formed in the fiber laser [24]. Figure 2c shows the corresponding polarization resolved optical spectra of the laser emission. The two spectra nearly overlap, indicating that they were coherently coupled. The two symmetric broad sidebands located at 1593 and 1599 nm on the spectra were caused by the periodic power variation of the laser beam in the cavity [27]. They are not the Kelly sidebands [28]. When increasing the pump power to 36.8 dBm, the widths of the dark-dark pulse pairs narrowed to about 600 ps. However, when continuing to increase the pump power, instead of the narrowing of the

pulse width continuing, the dark-dark pulse pairs started to split. New dark-dark pulse pairs with the same pulse width (about 600 ps) were generated, as shown in Figure 3. Due to the new pulse pair generation, the spacing between the pulse pairs became obviously unequal in the cavity.

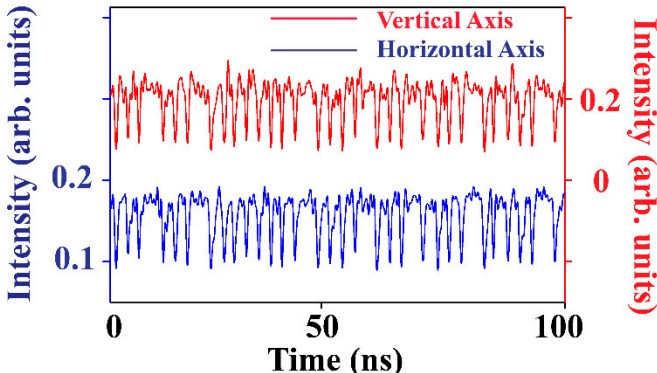

**Figure 3.** Splitting of the dark-dark pulse pairs under large cavity dispersion (EDF: 3 m; SMF: 0 m; DCF: 38 m).

In a previous paper, we reported the observation of vector dark-dark soliton formation in a fiber laser with a net normal average cavity GVD coefficient $\beta_2 \approx 0.61$ ps$^2$/km [29]. Although the dark solitons reported were incoherently coupled, their pulse widths were much narrower. Obviously, the dark-dark pulse pairs shown in Figures 2 and 3 are not a solitary wave in the sense of conventional dark solitons. We suspect this could be due to the fiber laser having a too large average cavity GVD. As soliton formation in media with large dispersion demands a higher threshold, if the cavity dispersion is too large, it is highly possible that the experimentally accessible pump power cannot reach the soliton formation threshold. To verify this, we inserted a piece of standard single-mode fiber (15 m, SMF-28) into the cavity, and shifted the average cavity GVD coefficient to $\beta_2 \approx 0.61$ ps$^2$/km. Although the laser then had a dispersion-managed cavity, under coherent cross polarization coupling, exactly the same feature as described above could still be observed, e.g., an in-phase periodic intensity modulation state, as shown in Figure 2a, could still be obtained in the laser at a much lower pump power. As the pump power was increased to 36 dBm, an in-phase periodic intensity modulation state could indeed evolve into a dark-dark vector soliton emission state, as shown in Figure 4. We note that, in Figure 4, the displayed dark pulse widths are limited by the bandwidth of our detection system.

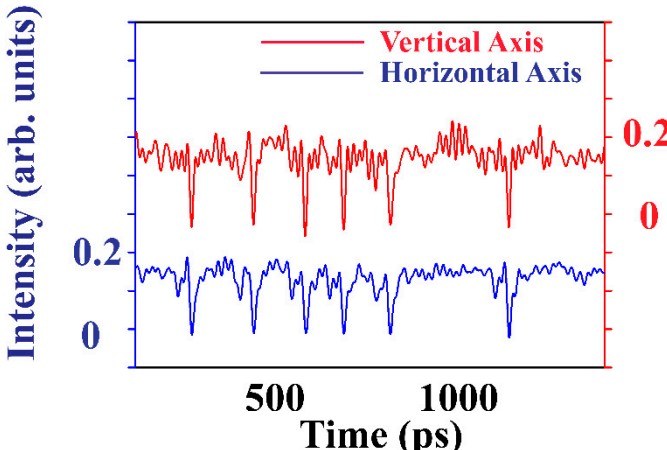

**Figure 4.** Coherently coupled dark-dark solitons under small net cavity dispersion (EDF: 3 m; SMF: 15 m; DCF: 38 m).

With a dispersion-managed cavity, the net average cavity dispersion could also be easily tuned to the anomalous dispersion regime. Furthermore, under coherent cross-polarization coupling, in-phase low frequency periodic intensity modulation could also be obtained in the fiber lasers. In contrast to fiber lasers with net normal average cavity dispersion where dark-dark vector solitons are obtained, such an in-phase periodic intensity modulation state obtained in the anomalous dispersion fiber lasers would quickly evolve into a phase-locked bright-bright vector soliton emission state, as reported in [22].

Under coherent cross-polarization coupling, not only in-phase periodic intensity modulation, but also anti-phase periodic intensity modulation, could be obtained. A typical case is shown in Figure 5a, where the polarization resolved laser emissions are anti-phase periodically modulated at a frequency of 333 MHz. Similar to the in-phase periodic intensity modulation cases described above, when we increased the pump power, the anti-phase intensity modulations evolved further into a train of dark-bright pulse pairs, as shown in Figure 5b. Figure 5c shows the corresponding polarization resolved optical spectra. We note that the bright pulses shown in Figure 5b are on a CW background; meanwhile, the intensities of the dark pulses do not drop to zero. Zhang et al. previously reported the observation of phase-locked dark-bright polarization domain wall solitons in a fiber laser [23]. Although both phenomena were formed as a result of coherent cross-polarization coupling between the two orthogonal oscillating laser cavity modes, they displayed different features. In the case of Zhang et al., domain wall solitons were formed, whose intensity always dropped to zero intensity. Again, limited by the large cavity dispersion, the dark-bright pulse pairs were not shaped into dark-bright vector solitons. To obtain dark-bright vector solitons, we further reduced the average cavity GVD coefficient using the cavity dispersion-management technique. In a cavity with an average GVD coefficient $\beta_2 \approx 0.96 \text{ps}^2/\text{km}$, the result shown in Figure 6 was obtained. Figure 6b presents the corresponding optical spectra. Both the Kelly sidebands and Four-wave mixing (FWM) sidebands confirmed that they were vector solitons. In addition to the CW background underneath the bright pulses disappearing, the depth of the dark pulses also reduced to zero intensity, which is a typical characteristic of coherently coupled black-white vector solitons [24].

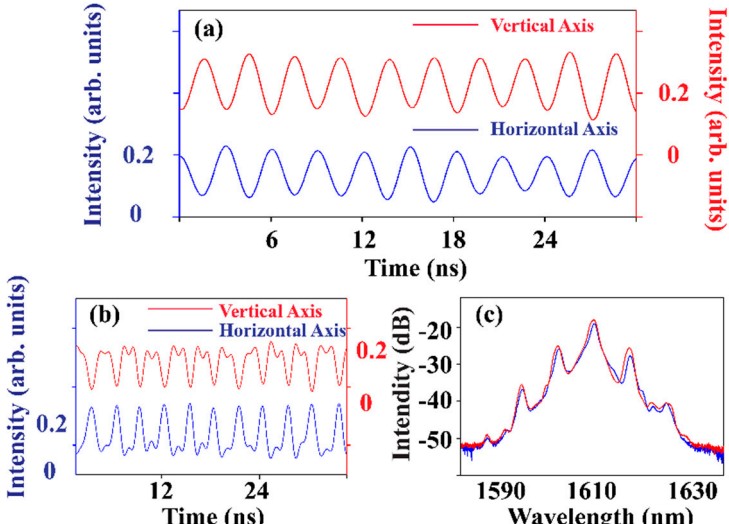

**Figure 5.** Evolution of periodic anti-phase intensity modulations in the fiber laser (EDF: 3 m; SMF: 0 m; DCF: 38 m). (**a**) Initial anti-phase periodic intensity modulation; (**b**) dark-bright pulse pairs formed; (**c**) the corresponding optical spectra of (**b**).

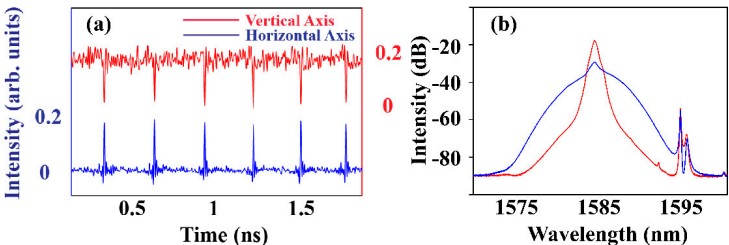

**Figure 6.** (**a**) Black-white solitons obtained under a net cavity group velocity dispersion (GVD) coefficient $\beta_2 \approx 0.99$ ps$^2$/km (EDF: 3 m; SMF: 9 m; DCF: 9 m). (**b**) The polarization resolved optical spectra.

Low frequency anti-phase periodic intensity modulations could also be obtained in the anomalous cavity dispersion regime. Again, in fiber lasers with a small average cavity GVD coefficient, the coherently coupled dark-bright vector solitons were experimentally obtained, suggesting that the occurrence of dark-bright vector solitons is independent of the sign of the cavity dispersion. However, in fiber lasers with large average cavity GVD coefficients, only coupled bright soliton-dark pulse pairs could be obtained [30].

## 3. Numerical Simulations

To corroborate the experimental observations, we numerically studied evolutions of periodic intensity modulations in a quasi-isotropic cavity single-mode fiber laser. In order to make the simulation results comparable to the experimental observations, we used the real fiber parameters in the simulations. Specifically, the fiber laser was made of two different types of fiber: One was an erbium-doped fiber (EDF) with a GVD coefficient of 61.87 ps$^2$/km, and the other was a dispersion compensation fiber (DCF) with a GVD coefficient of 5.1 ps$^2$/km. We assumed that the birefringence axes of the fibers were aligned and the laser cavity had sufficiently small linear birefringence. Therefore, the two orthogonal polarization components of the light in the laser were coherently coupled. We employed the technique described in [26] to numerically simulate the laser operation. Briefly, we always started a simulation with a weak initial CW field whose intensities along the two orthogonal polarization modes of the cavity were either in-phase or anti-phase periodically modulated. We let the light circulate unidirectionally in the fiber laser cavity. The light propagation in a weakly birefringent gain fiber is governed by coherently coupled Ginzburg–Landau equations (GLEs) [31]:

$$\frac{\partial u}{\partial z} = iku - \delta_1 \frac{\partial u}{\partial t} - i\frac{g}{\Omega_g}\frac{\delta}{(1+\delta^2)^2}\frac{\partial u}{\partial t} - i\frac{\beta_{2,eff}}{2}\frac{\partial^2 u}{\partial t^2} + i\gamma(|u|^2 + \frac{2}{3}|v|^2)u + \frac{i\gamma}{3}v^2u^* + \frac{g}{2}\frac{1+i\delta}{1+\delta^2}u$$
$$\frac{\partial v}{\partial z} = -ikv + \delta_1 \frac{\partial v}{\partial t} - i\frac{g}{\Omega_g}\frac{\delta}{(1+\delta^2)^2}\frac{\partial v}{\partial t} - i\frac{\beta_{2,eff}}{2}\frac{\partial^2 v}{\partial t^2} + i\gamma(|v|^2 + \frac{2}{3}|u|^2)v + \frac{i\gamma}{3}u^2v^* + \frac{g}{2}\frac{1+i\delta}{1+\delta^2}v \tag{1}$$

where $u$ and $v$ are the normalized envelopes of the optical fields along the two orthogonal polarized modes, $2k = 2\pi\Delta n/\lambda$ is the wave number difference between the modes and the inverse group velocity is represented by $2\delta_1 = 2k\lambda/2\pi c$, and $\beta_{2,eff}$ is the effective second order dispersion coefficient along the two orthogonal axes and can be represented by

$$\beta_{2,eff} = \beta_2 + \frac{g}{\Omega_g^2}\left[\frac{\delta(\delta^2 - 3) + i(1 - 3\delta^2)}{(1 + \delta^2)^3}\right], \tag{2}$$

where $\beta_2$ is the second order dispersion coefficient with the unit of $ps^2/km$, $\Omega_g$ is the gain bandwidth, $\delta = (\omega_0 - \omega_a)/\Omega_g$ is the gain detuning parameter, $\omega_0$ is the angular frequency of the propagating light

field, $\omega_a$ is the angular frequency of the fiber gain peak position, $\gamma$ represents the nonlinearity of the fiber, and g is the saturable gain coefficient of the gain fiber where

$$g = g_0 \exp\left[-\frac{\int \left(|u|^2 + |v|^2\right)dt}{E_{sat}}\right], \tag{3}$$

where $g_0$ is the small signal gain coefficient and $E_{sat}$ is the saturation energy. In the passive fiber, we set $g = \delta = 0$. When the light meets the cavity output port, 10% of the light intensity is deducted from the light fields, and the rest of the light fields are then re-injected into the cavity as the input for a new round of cavity circulation. We used the standard split-step Fourier method to solve the coupled extended GLEs (1). The numerical calculations were made on a 2 ns window and the periodic boundary condition was adopted. Using the simulation technique, many experimentally observed soliton features, such as soliton energy quantization [32] and noise-like pulse emission [33], are reproduced. In the current simulations, we focused on the evolution of periodic intensity modulations in fiber lasers under general laser operation conditions.

### 3.1. Evolution of In-Phase Intensity Modulations

We first investigated the situation where the initial periodic intensity modulation between the two orthogonal polarization components was in-phase and the laser cavity had a relatively large normal GVD coefficient. We used the following set of fiber parameters to simulate the laser operation: EDF fiber gain bandwidth $\Omega_g = 10\ nm$; small signal gain coefficient $g_0 = 60\ km^{-1}$; and gain saturation energy $E_{sat} = 1$ pJ. The fiber laser cavity had a 3 m EDF with a GVD coefficient of 61.87 ps$^2$/km, and a 38 m dispersion compensation fiber (DCF) with a GVD coefficient of 5.1 ps$^2$/km. Therefore, the average GVD coefficient of the cavity was $\beta_2 = 9.2$ ps$^2$/km. The beat lengths of the EDF and DCF were set to 50 km, so the average group velocity mismatch was $\delta_1 = 5 \times 10^{-5}$ ps/km. We assumed that the central wavelength of the light and the gain peak of the EDF w not matched, e.g., the central wavelength of the light was located at 1559.6 nm, while the peak gain of the EDF occurred at 1560 nm. Therefore, the gain detuning parameter was $\delta = 1.2$. The initial slowly varying envelopes of the orthogonally polarized optical fields were set as $u = A \cdot \sqrt{1 + (B \cdot \cos(\omega_c \cdot t))^2}$ and $v = A \cdot \sqrt{1 + (B \cdot \cos(\omega_c \cdot t))^2}$. Here, $A$ is related to the power level of the continuous wave background, $B$ is related to the intensity modulation depth, and $\omega_c$ is the angular frequency of the modulation. We circulated the light fields in the ring cavity until a stable operation state was obtained, and then considered the state obtained as a possible laser emission.

We set the modulation frequency at 5 GHz. A typical result obtained is shown in Figure 7. In that case, an initial in-phase periodic intensity modulation evolved to form a stable train of dark-dark pulse pairs. When increasing the pump power, which corresponds to increasing the $g_0$ and $E_s$ values, the CW power level of the dark pulses increased and the pulse width of the pulses decreased. The formed dark-dark pulse pairs were very stable in the cavity. However, as the pulse width narrowed to about 100 ps, further increasing the pump power, the dark-dark pulse pairs no longer narrowed; instead, they started to split up. The newly formed dark pulses also had a pulse width of about 100 ps, which is still much broader than those of conventional dark solitons formed in the laser [21]. The stable dark pulse train formation and dark pulse splitting under strong pumping are in good agreement with the experimental observation shown in Figure 3.

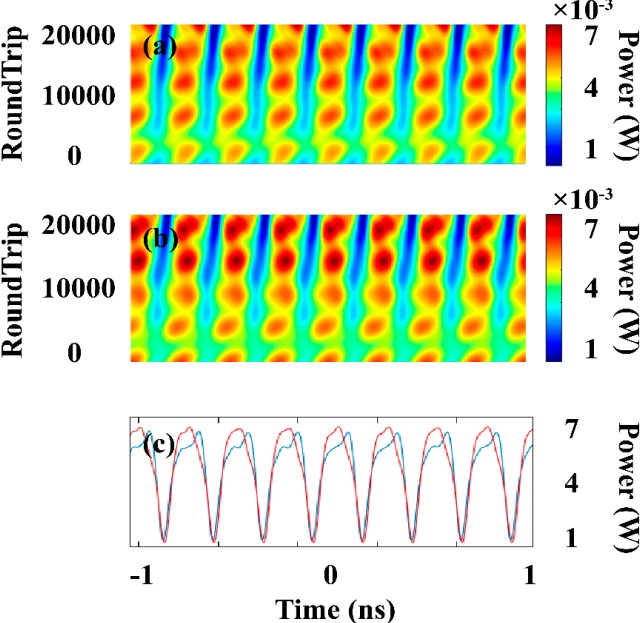

**Figure 7.** Evolution of initially in-phase intensity modulated vector fields to a train of dark-dark pulse pairs (EDF: 3 m; SMF: 0 m; DCF: 38 m). $A = 0.1$; $B = 1$; $\omega_c = 2\pi f_c$; $f_c = 5$ *GHz*. (**a,b**) Evolution of the in-phase intensity modulations along the two orthogonally polarized directions with the cavity roundtrips and (**c**) a stable train of dark-dark pulse pairs formed at the roundtrip of 20,000.

Numerically we found that, in order to form a stable train of dark-dark pulse pairs, the gain detuning parameter and laser gain bandwidth must be appropriately selected. Only in a certain range of gain detuning, i.e., $1 < \delta < 1.5$, and laser gain bandwidth, i.e., $10$ *nm* $< \Omega_g < 20$ *nm*, could the initial periodic intensity modulations evolve into a stable train of dark-dark pulse pairs. If the gain detuning effect was ignored, i.e., by setting $\omega_a = \omega_0$ and $\delta = 0$, no stable train of coupled dark-dark pulse pairs could be obtained, as shown in Figure 8.

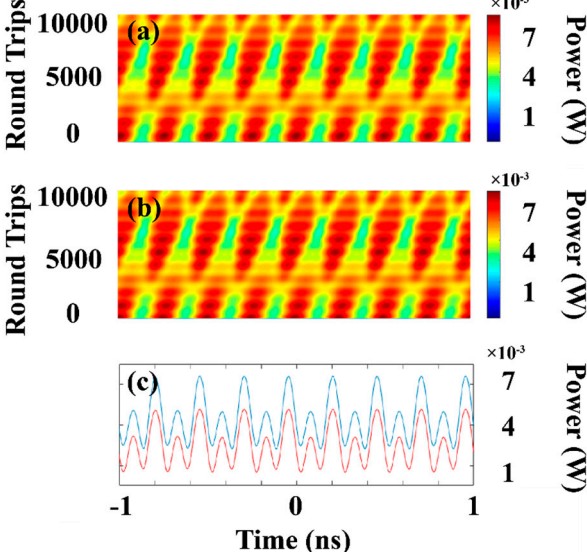

**Figure 8.** The same as in Figure 7, except for the gain detuning parameter $\delta = 0$. (**a,b**) Evolution of the in-phase intensity modulations along the two orthogonally polarized directions with the cavity roundtrips and (**c**) a state formed at the roundtrip of 10,000.

Experimental studies have shown that by implementing the cavity dispersion-management technique, the average cavity dispersion can be deliberately decreased. Therefore, with an experimentally accessible pump power, vector dark solitons can be formed from an in-phase intensity modulation state. We also numerically simulated the situation. To this end, we used a fiber laser whose cavity was made of a 3 m EDF, 9 m SMF, and 6 m DCF. The average GVD coefficient of the cavity was $\beta_2 = 0.43$ ps$^2$/km. We kept the other simulation parameters the same as those used in Figure 7. The result is presented in Figure 9. The initial in-phase intensity modulation was shaped into a stable train of vector dark solitons whose pulse width was about 8 ps.

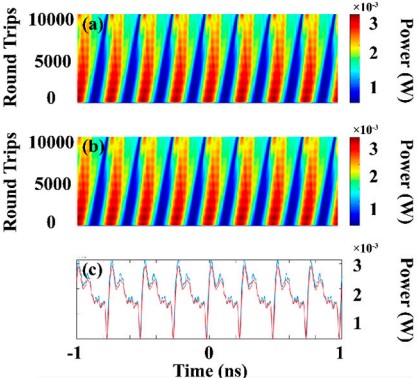

**Figure 9.** Formation of vector dark solitons from in-phase intensity modulations (EDF: 3 m; SMF: 9 m; DCF: 6 m). Parameters are the same as those used in Figure 7, except for the GVD coefficient $\beta_2 = 0.42$ ps$^2$/km. (**a,b**) Evolution of the in-phase intensity modulations along the two orthogonally polarized directions with the cavity roundtrips and (**c**) Vector dark solitons formed at the roundtrip of 10,000.

We also numerically investigated the evolution of in-phase intensity modulations in fiber lasers with anomalous cavity dispersion. As experimentally observed, in the anomalous dispersion cavity fiber lasers, bright-bright vector solitons were always formed, as shown in Figure 10. The bright solitons were equally spaced in the cavity, forming a so-called harmonic mode-locked vector soliton state. However, we note that no mode locking actually occurred in the fiber laser.

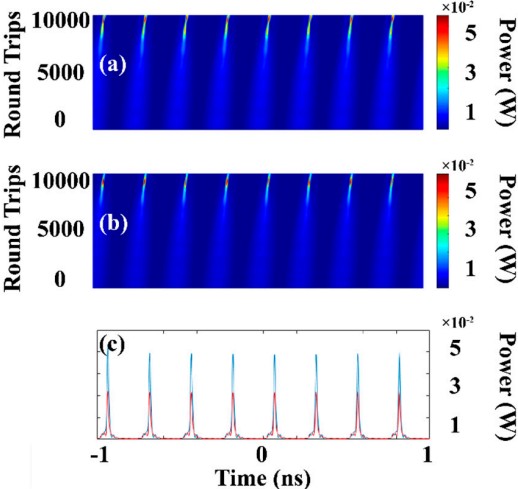

**Figure 10.** Formation of vector bright solitons from in-phase intensity modulations (EDF: 3 m; SMF: 10.7 m; DCF: 6 m). Parameters are the same as those used in Figure 7, except for the GVD coefficient $\beta_2 = -1.6$ ps$^2$/km. (**a,b**) Evolution of the in-phase intensity modulations along the two orthogonally polarized directions with the cavity roundtrips and (**c**) a stable train of vector bright solitons formed at the roundtrip of 10,000.

### 3.2. Evolution of Anti-Phase Intensity Modulations

To simulate the evolution of an initially cross-polarization anti-phase periodic intensity modulated light in the laser, we set the initial slowly varying envelopes of the optical fields as $u = A \cdot \sqrt{1 + (B \cdot \cos(\omega_c \cdot t))^2}$ and $v = A \cdot \sqrt{1 + (B \cdot \sin(\omega_c \cdot t))^2}$. The other simulation parameters were kept the same as those in Figure 7. Again, we circulated the light in the ring cavity for 10,000 roundtrips. The evolution of the light with the cavity roundtrips and the final stable state obtained are shown in Figure 11.

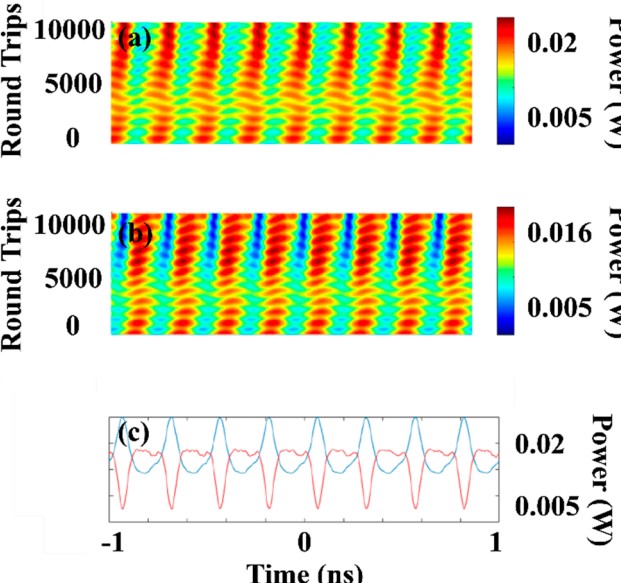

**Figure 11.** The same as Figure 7, except for the anti-phase intensity modulation. (**a**,**b**) Evolution of the anti-phase intensity modulations along the two orthogonally polarized directions with the cavity roundtrips and (**c**) a stable train of dark-bright pulse pairs formed at the roundtrip of 10,000.

Like the experimental observations, in cavities with a large normal GVD coefficient, the anti-phase intensity modulations evolved into a stable train of dark-bright pulse pairs. With the current parameter selection, the stable pulses had a width of about 100 ps. Similar to the case with in-phase periodic intensity modulations, in order to form a stable train of dark-bright pulse pairs, the gain detuning and laser gain bandwidth had to be carefully selected. The pulse width narrowed as the pump power increased. Once the pulse width narrowed to ~80 ps, further increasing the pump power, the dark-bright pulses started to break up.

We also simulated anti-phase periodic intensity modulations in dispersion-managed cavity fiber lasers where small net cavity dispersion could be realized. We numerically confirmed that in fiber lasers with a small average GVD coefficient, coherently coupled dark-bright vector solitons could be easily formed, either in the net normal or net anomalous cavity dispersion regimes, as shown in Figure 12, which is in good agreement with the experimental observations. Moreover, we also numerically verified that when a laser cavity displays relatively large net anomalous dispersion, e.g., $\beta_2 = -1.6$ ps²/km, starting from an initial anti-phase periodic intensity modulation state, a stable train of coupled bright soliton-dark pulse can also be obtained. Such a laser emission state was reported in [25].

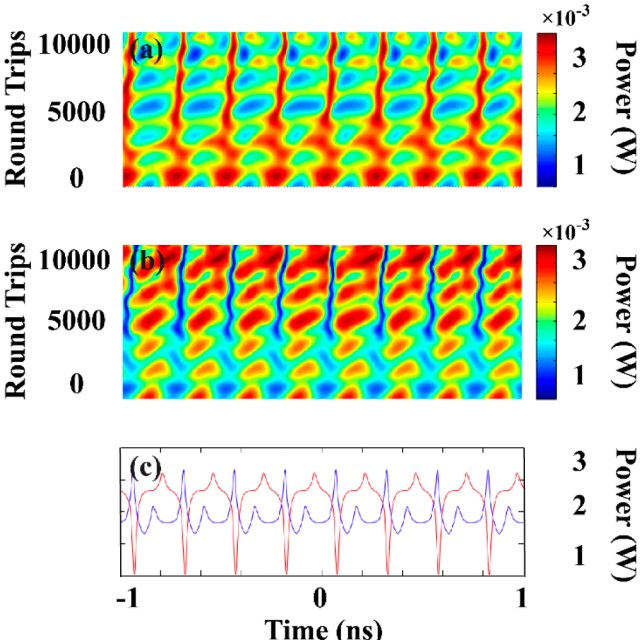

**Figure 12.** The same as Figure 11, except for the small average normal GVD coefficient $\beta_2 = 0.42$ ps$^2$/km (EDF: 3 m; SMF: 9 m; DCF: 6 m). (**a**,**b**) Evolution of the anti-phase intensity modulations along the two orthogonally polarized directions with the cavity roundtrips and (**c**) a stable train of dark-bright solitons formed at the roundtrip of 10,000.

## 4. Discussion

Our study focused on a coherent cross-polarization coupling case. Experimentally, this occurs when the net cavity birefringence of a fiber laser is sufficiently small. Our experiments reveal that, under coherent cross-polarization coupling, the laser emissions along the two orthogonal polarization directions of the cavity can become either in-phase or anti-phase periodically intensity modulated at a low frequency. The occurrence of these low frequency intensity modulations played a crucial role in the various types of vector solitons in our fiber lasers. Previous studies have shown that both the cross-gain saturation and PMI effects can lead to low intensity modulations [14,34,35]. However, the population relaxation time for an erbium dope fiber is particularly long, being about 10 ms. The slow intensity modulations observed in our experiments are in the order of hundreds of MHz. Therefore, we could exclude the possibility that the observed low frequency anti-phase intensity modulation was caused by cross-gain competition in the laser. Given the coherent cross-polarization coupling requirement and low frequency modulation feature, which are well-aligned with the theoretical prediction on the PMI in fibers [14,19], we would attribute the formation of the low frequency laser intensity modulations to the PMI in the fiber laser. Based on this understanding, one could also explain why the various vector solitons could be formed in the fiber laser, even without mode locking. We note that in the net anomalous dispersion regime, conventional MI also occurred in our fiber lasers. However, as the modulation frequency was very high, and is normally larger than 100 GHz, it could not be detected by our detection system. Moreover, we emphasize that, even under the existence of the MI effect in the anomalous dispersion regime, dark-bright vector solitons could still be formed in the fiber lasers and remain stable. This is different from the case of the light propagation in weakly birefringent SMFs [36], where the MI could destroy the dark-bright vector solitons.

Based on the coupled extended Ginzburg–Landau equations that also take into account the gain detuning effect, vector soliton shaping of the periodic intensity modulations was reasonably reproduced. We point out that, when obtaining a stationary and stable train of coherently coupled dark-dark or dark-bright pulse pairs in the simulation window, the gain detuning and gain bandwidth limitation play a crucial role. We suspect that this is due to the gain dispersion, as the laser could

always self-select a wavelength to fulfil the cavity boundary condition. Therefore, a train of stationary dark-dark or dark-bright vector solitons could be formed in the cavity.

Both the experimental studies and numerical simulations have shown that under in-phase intensity modulation, only dark-dark (bright-bright) vector solitons can be formed in normal (anomalous) dispersion cavity fiber lasers, while under anti-phase intensity modulations, independent of the sign of the cavity dispersion, dark-bright vector solitons can be formed. The formation of dark-bright vector solitons is a very interesting phenomenon. In contrast to dark-dark or bright-bright vector solitons, whose formation can also be considered as a result of the soliton–solitons interaction, the formation of coupled dark-bright vector solitons is purely a result of coherent cross-polarization coupling of the light in the laser. As the bright (dark) soliton formation is an intrinsic feature of the self-phase modulation in anomalous (normal) dispersion SMFs, dark-bright vector soliton formation can be considered an intrinsic feature of cross-phase modulation in SMFs.

Based on our research, coherently coupled dark-dark and dark-bright vector solitons can be relatively easily formed in fiber lasers with a small average cavity GVD. This could be related to the lower soliton formation threshold in the smaller GVD, as in the case of scalar soliton formation. To help readers understand our results, we have summarized the laser operation under different average cavity dispersions in Figure 13. Based on our research, we can now explain the physical mechanisms of the different laser operation states observed. Under weak net cavity birefringence, coherent cross-polarization coupling can lead to either in-phase or anti-phase low frequency periodic intensity modulations on the two orthogonal polarization components of the laser. Depending on the strength of the cross-polarization coupling, these in-phase or anti-phase low frequency periodic intensity modulations can then be shaped into dissipative dark-dark, bright-bright, or dark-bright pulse pairs or vector solitons.

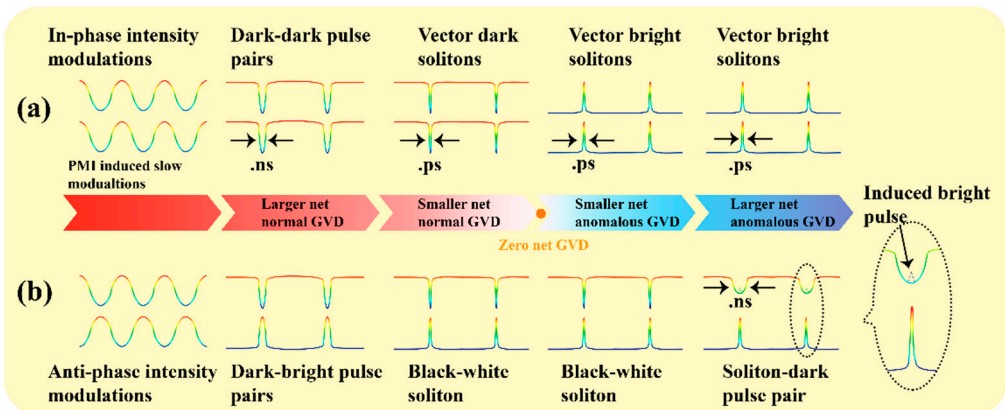

**Figure 13.** Vector dissipative solitons formed in different net cavity GVD. (**a**) Evolution of in-phase intensity modulations and (**b**) evolution of anti-phase intensity modulations.

## 5. Conclusions

In conclusion, we have presented detailed studies on the operation of a quasi-vector cavity fiber laser. Experimentally, we found that under coherent cross-polarization coupling, the intensity of the laser emissions along the two orthogonal polarization directions of the cavity could become either in-phase or anti-phase low frequency periodically modulated. Depending on the laser operation conditions, these low frequency periodic intensity modulations could further be shaped into dark-dark or dark-bright dissipative vector solitons in the fiber lasers. Numerical simulations also confirmed the experimental observations. Our research explained why, in a quasi-vector cavity fiber laser, even without any real or artificial saturable absorber in the cavity, different types of dissipative vector solitons can still be formed, as well as their formation conditions. Our results can provide a deep understanding of the vector soliton formation that occurs in fiber lasers.

**Author Contributions:** Conceptualization, D.T.; methodology, J.G. and X.H.; validation, X.H.; investigation, X.H.; data curation, X.H. and J.G.; writing—original draft preparation, X.H. and L.L.; writing—review and editing, J.G. and D.T.; supervision, D.T. and S.Y.; project administration, G.J. and D.T.; funding acquisition, D.T. and J.G. All authors have read and agreed to the published version of the manuscript.

**Funding:** This research received no extremal funding.

**Acknowledgments:** This work is supported partially by the National Natural Science Foundation of China (NSFC) (61875078,11704259); Priority Academic Program Development of Jiangsu High Education Institutions (PAPD); Minister of Education (MOE), Singapore (2018-T1-001-145); and A*Star AME IRG project (Project no. A1883c0003).

**Conflicts of Interest:** The authors declare no conflict of interest.

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
