# Peer review of "Evolution from Periodic Intensity Modulations to Dissipative Vector Solitons in A Single-Mode Fiber Laser"

_photonics, doi:10.3390/photonics7040103_

Round 1
Reviewer 1 Report
The authors report experimental and theoretical results on bright-bright, dark-dark or dark-bright vector solitons in a fiber laser. The research explains why different types of dissipative vector solitons can be formed in a quasi-vector cavity fiber laser even without a saturable absorber in a cavity. Formation conditions for different types of dissipative vector solitons are found and discussed. The work is extensive and scientifically significant. It is a continuation of a series of works by the authors related to the study of vector solitons. New results are reported here and they seem to be reliable and well-founded.
I can recommend the authors to pay attention to the following points.
- In Line 212, the authors say that they set about 10000 roundtrips in the simulation. It should be explained how one can choose the proper number of roundtrips. What quantitative criteria are established? It is not obvious from Figures 7a,7b, 8a,8b,9a,9b etc that the regimes are stationary (maybe if one waits another 20000 roundtrips, the pattern will change?)
- How many points in the spectral window are taken and how long does the calculation take?
- What happens if one sets the initial conditions in the form of noise, as in a real laser (and not as, for example, in lines 209,210)?
- When the average cavity GVD coefficient is close to zero, is the beta3 coefficient affected (in line 150 beta2 = 0.96 ps^2/km, the corresponding results are shown in Fig. 6)?
Formatting:
In the article, there are often dots after the word "Figure" (it should be deleted), for example, in line 59 (Figure. 1. -> Figure 1). The similar in lines 83,91,94,102,107 etc.
Measurement units are often written in non-standard font (for example, in lines 113, 150, 184, 201,202, etc).
Author Response
Dear reviewers,
The reply letter is attached.

Reviewer 2 Report
The authors have theoretically and experimentally studied the evolution of periodic intensity modulations to dissipative vector solitons in a single mode fiber laser. The paper is clear but the novelty is limited (many publication by the group, eg., Hu etal, Phys. Rev. A 101, 063807, 2020). The following points must be fully addressed considering a major revision.
The literature review is weak and more references must be added including the MI not only in fibers but EDFA and lasers considering higher order dispersions; nonlinearity terms and so on, e.g.,
Agrawal, G. P., Modulation instability in erbium-doped fiber amplifiers, IEEE Photon. Technol. Lett., 4, 562-564 (1992).
Tehranchi, A. etal., Induced modulational instability in EDFAs in the presence of higher-order nonlinear and dispersive effects. J. of Opt. and Quantum Electron. 39, 651-658 (2007).
Luo, Z., etal., Modulation instability induced by periodic power variation in soliton fiber ring lasers. Eur. Phys. J. D 54, 693–697 (2009).
Luo, Z., etal., Modulation instability induced by cross-phase modulation in a dual-wavelength dispersion-managed soliton fiber ring laser. Appl. Phys. B 100, 811–820 (2010).
Peng. J. etal., Experimental observation of transitions of different pulse solutions of the Ginzburg-Landau equation in a mode-locked fiber laser. Phys. Rev. A 86, 033808 (2012).
The choice of DCF length is not clear. Random? Is it supposed to increase dispersion in the loop…Is there any SMF in the loop? What are the connectors? How much are the fibers nonlinearities? The EDFA parameters?
Are the generated pulses stable in exp.? Why are they not shown in a bigger time span?
How much is the pump bandwidth? Any Brillouin generation?
How much is the input pump power in exp.?
How much is the input/output power for different cases?!
What about Raman generation?
How do they ensure that the net cavity birefringence is sufficiently small?
How do they resolve vertical and horizontal axis, in practice? Explain.
How do they monitor the laser emission spectra along the two orthogonal polarization directions?
Why the time scale is so different in Figs. 3 and 4? Any justification for pulse widths? The captions are too short and non-informative.
In simulation, why do they not consider Raman and other nonlinear terms in Eqs. 1? For small beta_2 why do they not consider beta_3? What is Eqs.’ Reference?
Why is the modulation frequency set at 5GHz in the simulation? (vs. the theory)
In line 213, they claim: (…consider the steady state obtained as a possible laser emission.). What is a steady state?! Can you show this steadiness? Why at 10000 RTs. Explain. In fact, contour figures must be shown for more than 10000.
Fig.6b. What are the side lobes?
Fig. 7. What is the difference between a and b?
Line 227: What feature does well agree? Pulse width? Repetition?
Fig. 13. What is black-white? No description in the text.
Author Response
Dear reviewer,
The reply letter is attached.
Thanks.

Round 2
Reviewer 2 Report
The manuscript has much improved and deserves publication after small corrections.
-The “final stable state” is still not an accurate phrase for your choice of RT. The pulses at other RTs might be also at stable state, eg., from the point of peak intensity. It is obvious that they seem almost periodic over RTs and your choice could ONLY be a selective state (eg., with maximum intensity) resembling the experimental results.
-Modification of Fig. 1 after new details is recommended.
Author Response
Dear Reviewer,
The response letter is attached.
Thanks.
